# Spatiotemporal differentiation and analysis of factors influencing high-quality development of resource-based cities: An empirical study based in China

**Jun Liu**[1], **Changchun Lin**[1], **Xin Wang**[2], **Xiuli Liu**[2]*

**1** Department of Geography, Xinzhou Normal University, Xinzhou, China, **2** Resource-Based Economy Transformation and Development Institute, Shanxi University of Finance & Economics, Taiyuan, China

* lxl820113@163.com

**Data Availability Statement:** All relevant data are within the manuscript and its Supporting information files.

## Abstract

Resource-based cities often face problems such as resource scarcity and insufficient electricity to achieve complex high-quality growth. At present, there is relatively little research on the impact on the high-quality development of such cities. To study the key variables that affect the high-quality growth of resource-based cities, we adopt entropy weighted TOPSIS technology, spatial correlation analysis, and spatial econometric models. The main conclusions are as follows: (1) The overall high-quality development of resource-based cities in China is on the rise year by year; The cities with the highest growth rates are those that are mature, rejuvenated, growing, and declining. (2) Resource-based cities have a positive geographical correlation in high-quality development, and different numbers of clusters are displayed by changing the Moran I index score. (3) High quality development is strongly influenced by human capital, urbanization, technological innovation, and global market openness. There are significant differences in the ways in which these variables affect various types of resource-based cities. Policy makers who strive to reduce regional inequality and encourage high-quality growth in resource-based communities may benefit greatly from the insights provided by this study.

## 1. Introduction

China's economy has shifted from high-speed growth to a stage of high-quality development [1]. This high-quality development is a crucial requirement for current and future regional development [2]. However, approximately 40% of China's cities are resource-based, serving as vital energy security bases that significantly contribute to the country's economic and social progress. However, owing to the non-renewable nature of resources, high-quality development of several resource-based cities can be challenging. They face problems resulting from single industrial structure, slow economic growth, insufficient innovation ability, a lack of energy resources, deterioration of the ecological environment, and increasing pressure on people's livelihoods [3, 4].

**Funding:** The authors received support from the National Natural Science Foundation of China (Grant Nos. 42001257 and 71874119), Philosophy and Social Sciences Research of Higher Learning Institutions of Shanxi (Grant No. 20210115), and the Shanxi Postgraduate Education Innovation Project (Grant No. 2021Y571). We also thank our anonymous reviewers and the editor for their helpful suggestions.

**Competing interests:** The authors have declared that no competing interests exist.

To address these challenges, China issued several opinions in 2007 to promote the sustainable development of resource-based cities, emphasizing the establishment and improvement of a long-term mechanism for development [5]. In 2013, the national sustainable development plan for resource-based cities (2013–2020) [6] proposed sustainable development paths for these cities. In 2017, further emphasis was placed on the transformation and development of resource-based cities, highlighting high-quality development aligned with the five development concepts of "innovation, coordination, green, openness, and sharing." The 19th CPC National Congress report pointed out that sufficient support should be provided for the transformation and development of resource-based cities. Therefore, promoting the high-quality development of resource-based cities is necessary, not only to solve their developmental problems but also to implement a national strategy.

Currently, research in academia on high-quality development mainly focuses on its connotation, meaning, and evaluation [7–9], and a unified evaluation index system for high-quality development has not yet been formed, especially for resource-based regions. There is a lack of research on the evaluation index system for high-quality development in resource-based regions. In addition, scholars often study resource-based cities at national, provincial, and urban levels, examining aspects like labor quality [10], environmental pollution [11, 12], and education quality [13], but research on factors influencing high-quality development remains limited.

To fill these gaps, our goal is to examine resource-based cities across China, systematically analyze the spatiotemporal evolution characteristics of high-quality development levels, and the impact mechanisms of different types of resource-based cities. This article takes 113 resource-based cities in China from 2003 to 2018 as samples, uses entropy weighted TOPSIS method and spatial correlation analysis to construct a high-quality development evaluation system, and studies the spatiotemporal evolution trends of high-quality development in different types of cities. Meanwhile, using spatial econometric models, this study explores the roles of industrial structure, technological innovation, human capital, urbanization, environmental regulation, openness, and infrastructure in high-quality development. Finally, based on empirical results, high-quality development suggestions were proposed.

The contribution of this study is mainly reflected in three aspects: firstly, this study developed an inclusive and widely applicable evaluation index system for the high-quality development of resource-based cities. This innovative system is based on the "Five Development Concepts" and fills an important gap in the current research field, providing a comprehensive framework for evaluation. Secondly, by incorporating most resource-based cities in China into the research sample, this study provides a comprehensive analysis with practical significance covering a wider range of regions. This helps to improve the applicability of research results. Finally, this study selected relevant influencing factors through scientific and targeted methods, thus filling the research gap in the factors affecting the high-quality development of resource-based cities. Through this comprehensive analysis, this study reveals the challenges faced by resource-based cities in the process of high-quality development and provides wise solutions, which helps decision-makers better understand the essence of high-quality development in resource-based cities.

The rest of this study is structured as follows. Part 2 introduces the relevant literature on high-quality development in resource-based cities. Part 3 describes the research methods, study areas, data sources, and variable selection. Part 4 presents the research structure and analysis, including spatio-temporal evolution analysis of high-quality development in resource-based cities and empirical testing of influencing factors. Part 5 is the conclusion and policy implications of the study.

## 2. Literature review

The academic community's understanding of the connotation of high-quality development has undergone a transformation from focusing solely on the quality of economic growth to development quality. Initially, people's understanding of high-quality development was mostly focused on economic development, and they believed that when measuring regional development, using only per capita GDP was not comprehensive enough, and that people's lives and social welfare should also be included [14, 15]. As scholars have conducted more in-depth research on the connotation of high-quality development, they have gradually incorporated aspects such as ecological environment, society, and human capital into the evaluation index system for measuring regional high-quality development [16–18]. At this point, the academic community has reached a consensus on the definition of the connotation of high-quality development, generally believing that high-quality development is efficient and high-quality development that can meet people's material, social, and psychological needs.

Currently, with the proposal of the five development concepts of innovation, coordination, green, openness, and sharing, the connotation of high-quality development has been further deepened [19]. These concepts, proposed by the Communist Party of China in the new era, encompass innovation, coordination, green development, openness, and sharing, guiding China's economic, political, and cultural development towards high-quality outcomes [20]. Specifically, innovation refers to promoting technological and industrial innovation to improve the quality and efficiency of economic development; coordination refers to promoting coordinated development between urban and rural areas, industries, regions, and sectors; green development refers to promoting ecological civilization construction, protecting the environment, and achieving coordinated development between economic development and environmental protection; openness refers to deepening opening-up, actively participating in global economic governance, and promoting economic globalization; sharing refers to promoting social fairness and justice, strengthening social security, and achieving more equitable and inclusive distribution of development outcomes.

The Five Development Concepts are interconnected and mutually reinforcing, forming an integrated whole with inherent connections [21]. Compared with the previous connotation of high-quality development, the five development concepts have more systematic and comprehensive characteristics, more emphasis on people-oriented development, and are more in line with the trend of the times, providing more comprehensive theoretical support for evaluating regional development.

Regarding the evaluation of high-quality development, most scholars use evaluation methods such as the entropy weight method [11, 22–24] and TOPISI method [25, 26] to construct evaluation index systems and calculate the level of high-quality development, and a few scholars use methods such as DEA model [27] and coupling coordination degree analysis [28–30] to evaluate the development of regions with high quality. In addition, some scholars have conducted comparative studies on the relationship and differences between high-quality development and sustainable development, believing that early high-quality development was more focused on economic quality compared to sustainable development [31]. However, with the proposal of the five development concepts, the content and scope of high-quality development have continued to expand [32], leading to the gradual formation of a unified whole in which high-quality development and sustainable development are mutually coordinated and complementary.

The research on the mechanism of high-quality development of resource-based cities mainly focuses on institutional construction, financial support, and industrial structure. For

example, Al-Ubaydli [33] and Wiens [34] propose a high-quality institutional design to avoid the resource curse and foster transformation. Song Liying et al. [35] suggest adjusting fiscal autonomy and expanding special funds for resource-based cities based on fiscal autonomy's role in total factor productivity Jin Bei [36] believes that the government plays an important role in high-quality development, especially in the western region, which relies heavily on government support. At the same time, he suggests that technological innovation is a vital source of high-quality development. Yang et al. [37] emphasizes the challenges in single industrial structures, ecological neglect, and regulatory inefficiency hindering high-quality development. In addition, regarding the development path of resource-based cities, Yang et al. [5] points out that the transformation and development of resource-based cities cannot be achieved without long-term financial and policy support from the government. Li et al. [38] propose that resource-based cities should construct ecological environmental protection and ecological compensation mechanisms to coordinate sustainable development and ecological protection of resources.

## 3. Materials and methods

### 3.1 Research methods

**3.1.1 Measurement of high-quality development level of resource-based cities.** *(1) Indicator system.* This study is based on "five development concepts". It also draws on relevant research on sustainable development [39], human development evaluation [40], high-quality development in China in the new era [41–43], and resource-based city transformation [44]. According to the special objective situation of resource-based cities [44], this study establishes a high-quality development index system for resource-based cities. Six subsystem evaluations of economy, innovation, coordination, greenness, openness, and sharing are also constructed. Using the entropy weight method, the weight of each evaluation index is determined (Table 1).

*(2) Entropy-weighted TOPSIS method.* The entropy weighting method is an objective weighting method with a dimensionless approach adopted for data processing. This method effectively reflects the role of each index in the evaluation object and is more rigorous compared to subjective empowerment. The TOPSIS method is simple and it evaluates the relative distance between each measurement object and the best and worst of all schemes, and then quantifies the ranking. The entropy weight method is used to calculate the weight and rank the high-quality development level of each resource-based city. The calculation process is as follows.

(a) Standardization of the evaluation matrix

The standardized matrix is obtained using the range transformation method:

The processing of positive and negative correlation indicators is as follows:

$$R = \begin{bmatrix} x_{11} \ x_{12} \ \dots \ x_{1n} \\ x_{21} \ x_{22} \ \dots \ x_{2n} \\ \dots \ \dots \ \dots \ \dots \\ x_{m1} \ x_{m2} \ \dots \ x_{mn} \end{bmatrix} \tag{1}$$

$$x_{ij} = \frac{V_{ij} - \min(V_{ij})}{\max(V_{ij}) - \min(V_{ij})} \tag{2}$$

Table 1. Evaluation index system and weight of high-quality development of resource-based cities.

| Criterion layer | | | Indicator layer | Entropy Weight Method Weight | Indicator properties |
|---|---|---|---|---|---|
| High-quality development based on "five development concepts" | economy | growth efficiency | GDP/social workers | 0.025 | + |
| | | | GDP growth rate | 0.037 | + |
| | | | Growth rate of social fixed asset investment | 0.03 | + |
| | | growth level | Real GDP per capita | 0.029 | + |
| | | | per capita fiscal revenue | 0.034 | + |
| | | | gross industrial output per capita | 0.028 | + |
| | innovation | innovative research and development | Science and technology expenditure | 0.032 | + |
| | | | Number of granted patent applications | 0.023 | + |
| | | | The number of green invention patents | 0.029 | + |
| | | innovation potential | Education spending as a share of GDP | 0.03 | + |
| | | | Number of teachers per 10,000 people | 0.019 | + |
| | | transformation development | Energy consumption per ten thousand yuan of GDP | 0.025 | - |
| | | | Employees in the tertiary industry among 10,000 people | 0.045 | + |
| | coordination | regional coordination | Thiel Index | 0.031 | + |
| | | demand coordination | Total per capita consumption | 0.031 | + |
| | | | Engel's coefficient of urban residents | 0.024 | - |
| | | | Year-end urban registered unemployment rate | 0.032 | - |
| | | industrial structure | Primary and secondary industry coordination degree | 0.027 | - |
| | | | Primary and tertiary industry coordination degree | 0.027 | - |
| | | | Industrial development potential | 0.025 | + |
| | green | resource consumption | Energy consumption per capita | 0.021 | - |
| | | | Household water consumption per capita | 0.017 | + |
| | | ecological environment | Total industrial "three wastes" emissions per 10,000 yuan of industrial added value / t 10,000 yuan -1 | 0.017 | - |
| | | | PM2.5 | 0.02 | - |
| | | | Green coverage rate of built-up area | 0.025 | + |
| | | environmental governance | Comprehensive utilization rate of industrial solid waste | 0.017 | + |
| | | | Centralized treatment rate of sewage treatment plant | 0.022 | + |
| | | green living | Green area per capita | 0.028 | + |
| | | | Harmless treatment rate of domestic pollution | 0.023 | + |
| | open | trade open | Total import and export trade per capita | 0.02 | + |
| | | investment open | Utilize the total amount of foreign investment/ GDP | 0.034 | + |
| | | | Number of foreign-invested enterprises | 0.027 | + |
| | shared | public service | Per capita education expenditure of local finance | 0.031 | + |

$$x_{ij} = \frac{max\left(V_{ij}\right) - V_{ij}}{max\left(V_{ij}\right) - min\left(V_{ij}\right)} \tag{3}$$

where $x_{ij}$ refers to the data value of the $i$ year corresponding to the $j$ evaluation object; $i$ = 1, 2,

..., $n$ is the corresponding number of years in the evaluation system; $j = 1, 2, ..., m$ is the number of indicators.

(b) The weight of each index in the high-quality development evaluation index system is calculated by:

$$w_j = \frac{g_j}{\sum_{j=1}^{m} g_j} \tag{4}$$

where, for the $j$ index, the difference coefficient is $g_j = 1 - e_j$, $e_j \geq 0$; Information entropy is $e_j = -\frac{1}{ln(n)} \left( \sum_{i=1}^{n} P_{ij} \, ln^{P_{ij}} \right)$. In the system, the feature proportion occupied by the evaluation index is $P_{ij} = \frac{x_{ij}}{\sum_{i=1}^{n} x_{ij}}$. The greater the weight of the subsystem, the greater its contribution to high-quality development.

(c) Construction of a weighted matrix $R$ of high-quality development level measurement indicators:

$$R = \left( r_{ij} \right)_{n \times m} \tag{5}$$

where $r_{ij} = w_j \times x_{ij}$.

(d) Determination of the optimal scheme $Q_j^+$ and worst scheme $Q_j^-$ according to the weighting matrix $R$:

$$Q_j^{+(maxr_{i1}, maxr_{i2}, \cdots, maxr_{\jmath})} \tag{6}$$

$$Q_j^- = (minr_{i1}, minr_{i2}, \cdots, minr_{\jmath}) \tag{7}$$

(e) Calculation of the Euclidean distances $d_i^+$ and $d_i^-$ between each measurement method and the optimal scheme $Q_j^+$ and worst scheme $Q_j^-$:

$$d_i^+ = \sqrt{\sum_{j=1}^{m} \left( Q_j^+ - r_{ij} \right)^2} \tag{8}$$

$$d_i^- = \sqrt{\sum_{j=1}^{m} \left( Q_j^- - r_{ij} \right)^2} \tag{9}$$

(f) Calculation of the relative proximity $C_i$ between each measurement method and ideal solution:

$$C_i = \frac{d_i^-}{d_i^+ + d_i^-} \tag{10}$$

where the relative proximity $C_i$ is between 0 and 1; the larger the value, the better the level; otherwise, it is worse.

**3.1.2 Spatial autocorrelation model.** *(1) Global spatial autocorrelation model.* This study uses the spatial autocorrelation model to analyze the spatial evolution trend of the high-quality development of resource-based cities. Spatial autocorrelation includes both global and local autocorrelation. Global autocorrelation can be used to analyze the overall spatial correlation of resource-based cities. The formula for calculating the global Moran's I index is:

$$I = \frac{n(x_i - \bar{x}) \sum_{j=1}^{n} w_{ij} \left( x_j - \bar{x} \right)}{\sum_{i=1}^{n} \left( x_i - \bar{x} \right)^2} \tag{11}$$

where $x_i$ and $x_j$ represent the value of high-quality development level or efficiency, respectively, $n$ represents the total number of observations, $\bar{x}$ is the mean value of high-quality development, $w_{ij}$ is the spatial weight matrix, $i$ and $j$ represent two resource-based cities, the value of $I_i$ is [−1,1], $I_i = 0$ indicates no spatial autocorrelation, $I_i > 0$ indicates a positive correlation, and $I_i < 0$ a negative correlation. In this study, a spatial adjacency matrix is selected for the spatial weight matrix. If study areas $i$ and $j$ are adjacent, the value is 1, otherwise, it is 0.

*(2) Local spatial autocorrelation model.* As the global spatial autocorrelation Moran's I index cannot reflect the heterogeneity between regions, the local Moran's I index, namely the local index of spatial correlation (LISA), is introduced to reveal the relationship between resource-based and neighboring cities using ArcGIS 10.2 software. That is, there is a spatial correlation between high-quality development. The calculation formula is as follows:

$$I = \frac{n \sum_{i=1}^{n} \sum_{j=1}^{n} w_{ij}(x_i - \bar{x})\left(x_j - \bar{x}\right)}{\sum_{i=1}^{n} \sum_{j=1}^{n} w_{ij} \sum_{i=1}^{n} (x_i - \bar{x})^2} \tag{12}$$

where, for resource-based city $i$, $I_i$ is local Moran's I index. When it is greater than 0, it indicates a similar value in the spatial agglomeration of city I and adjacent areas. That is, it belongs to a high-high agglomeration area or a low-low agglomeration area. When it is less than 0, it indicates a different value in the spatial agglomeration, and the value of city I is opposite to that of the adjacent cities. The larger the absolute value of $I_i$, the higher the proximity.

**3.1.3 Spatial econometric model.** The high-quality development of resource-based cities is not only affected by these factors but also by the development of neighboring cities. Therefore, this study uses a spatial econometric model to conduct an empirical test of the factors affecting the high-quality development of resource-based cities using Stata 15.0 software. Among spatial econometric models, the spatial lag model (SAR), spatial error model (SEM), and spatial Durbin model (SDM) are more commonly used [11].

*(1) Model setting.* In the SAR model, there are endogenous interaction effects:

$$y_{it} = \alpha + \delta \sum_{j=1}^{n} w_{ij} y_{jt} + x_{it} \beta + \mu_i + \lambda_t + \varepsilon_{it} \tag{13}$$

In the SEM model, there is an interaction effect of the error term:

$$y_{it} = \alpha + x_{it} \beta + \mu_i + \lambda_t + \mu_{it} \tag{14}$$

$$\mu_{it} = \rho \sum_{j=1}^{n} w_{ij} \mu_{jt} + \varepsilon_{it} \tag{15}$$

In the SDM model, there are both endogenous and exogenous interaction effects:

$$y_{it} = \alpha + \delta \sum_{j=1}^{n} w_{ij} y_{jt} + x_{it} \beta + \sum_{j=1}^{n} w_{ij} x_{jt} \theta + \mu_i + \lambda_t + \varepsilon_{it} \tag{16}$$

where $\delta$ is the coefficient of spatial auto-regression, $w_{ij}$ is the spatial weight of all elements, $y_{it}$ is the explained variable, $x_{it}$ is the explanatory variable, the coefficient is $\beta$, $\theta$ is the coefficient of lag, $\rho$ is $\theta$; the coefficient of spatial autocorrelation is $\rho$; $\mu_i$ is the regional effect, $\mu_{it}$ is the spatial error term; $\lambda_t$ is the time effect, $\varepsilon_{it}$ is the random error term.

*(2) Variable selection.* This study considers the high-quality development level (Y) as the explained variable, selects the panel data of 113 resource-based prefecture-level cities in China from 2003–2018 as a sample, considers the availability of data, combines existing research [45–48], and selects the following explanatory variables:

Industrial structure level (IS): Most resource-based cities in China face an economic transformation dilemma. Regarding GDP, the added value of the secondary industry has a more

significant impact on the economy than does that of the primary and tertiary industries. Therefore, the industrial structure level of this study is represented by the proportion of the secondary industry's added value in GDP [44].

Scientific and technological innovation level (RD): Scientific and technological research and development is an important source of power to promote technological progress, improve innovation capabilities, and achieve economic growth. This study uses the proportion of science and technology expenditures in the GDP as a representative variable of urban support for innovation.

Human capital level (HR): The improvement of the human capital level can effectively improve the utilization rate of production factors, such as physical capital and technological research and development, and promote high-quality development. The level of human capital in this study is measured by the proportion of students in traditional institutions of higher learning in the population.

Level of Urbanization (UPN): The urbanization process results in a concentration of advanced production factors and resources in cities and a widened income gap between urban and rural areas. Conversely, the flow of population and resources between urban and rural areas is conducive to narrowing the urban-rural gap and radiating to drive rural development. In this study, the urbanization rate is a representative variable of the urbanization level [48].

Environmental regulation (ER): To achieve high-quality development of resource-based cities, it is necessary to reduce dependencies on resource-based industries and promote green and sustainable development [49]. Studies have shown that environmental regulation and high-quality growth are non-linear and that environmental regulation within a reasonable range has a positive impact on high-quality development [45]. In this study, environmental regulation is measured by the proportion of environmental pollution control investment in GDP.

Level of openness to the outside world (IEP): Open development is conducive to the optimal allocation of resources. The energy shortage of resource-based cities can be effectively alleviated through the unimpeded flow of factors inside and outside the region. Simultaneously, advanced production technologies and production of high- value-added products can also help. Industrial structures should be optimized and upgraded to improve the quality of economic growth [46]. The level of foreign trade in this study is expressed by the degree of dependence on foreign trade, that is, the proportion of each city's import and export volumes in the GDP [48].

Infrastructure level (IT): Economic development requires sound infrastructure that positively promotes economic growth. However, infrastructure construction costs are high and sometimes crowd out other investments. This study uses highway traffic density as a proxy variable for infrastructure, that is, the ratio of highway mileage to the administrative area.

Individual missing data are estimated using linear interpolation, and the data are logarithmically processed to eliminate the effect of heteroscedasticity. The data processing software used in this paper is Stata15.0. The descriptive statistics for the variables are presented in Table 2.

## 3.2 Data sources

This study selects 126 resource-based prefecture-level cities identified in the "National Sustainable Development Plan for Resource-based Cities (2013–2020)" [6]. However, Laiwu City was merged with Jinan City in 2019. The statistical data of Bijie, Pu'er, and Daxinganling Cities, and ethnic minority autonomous prefectures were substantially missing; therefore, only 113 resource-based prefecture-level cities are retained as the study areas, and the study period is 2003–2018. The reason for selecting the sample study period as 2003–2018 is that the basic establishment of China's socialist market economy system occurred after 2002, and the domestic institutional environment has become relatively stable. Additionally, China joined the

**Table 2. Descriptive statistics of variables.**

| Variable | Observation number | Average value | Standard deviation | Minimum value | Maximum value |
|---|---|---|---|---|---|
| Y | 1808 | 0.483 | 0.116 | 0.235 | 0.741 |
| IS | 1808 | 50.407 | 12.272 | 9.000 | 90.970 |
| RD | 1808 | 0.160 | 0.214 | 0.008 | 6.310 |
| HR | 1808 | 0.009 | 0.008 | 0.000 | 0.058 |
| UPN | 1808 | 47.290 | 14.985 | 11.574 | 100.000 |
| ER | 1808 | 0.973 | 1.268 | 0.001 | 22.335 |
| IEP | 1808 | 8.699 | 14.857 | 0.008 | 145.778 |
| IT | 1808 | 0.802 | 0.495 | 0.030 | 2.417 |

World Trade Organization at the end of 2001, so the external environment faced by China during the sample period was relatively consistent [50]. Furthermore, due to the outbreak of the COVID-19 pandemic in China in 2019, the data for 2019–2020 is relatively distorted, and some data for 2021–2022 has not yet been released.

The data are obtained from the "China Urban Statistical Yearbook" (2003–2019), the Statistical Yearbook of each province and city (2003–2019), the statistical bulletin of the national economic and social development of each city, and the government work report (2003–2018), combined with the "National Sustainable Development Plan for Resource-Based Cities (2013–2020)". The 113 resource-based prefecture-level cities are divided into the following four types (Fig 1).

# 4. Results

## 4.1 Analysis of spatiotemporal evolution of high-quality development of resource-based cities

**4.1.1 Measurement results of high-quality development of resource-based cities.** Based on the evaluation index system of high-quality development of resource-based cities, the

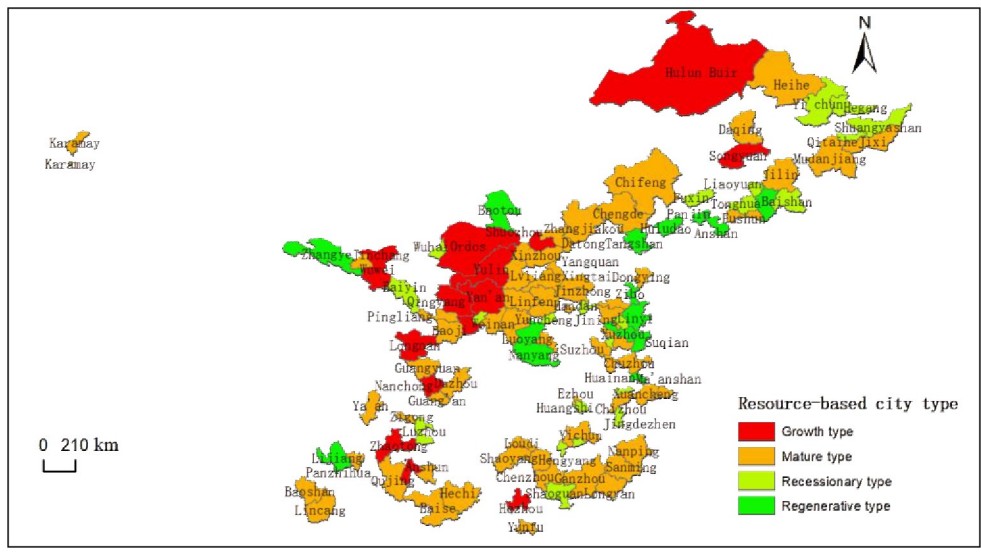

**Fig 1. Spatial distribution of resource-based cities.**

entropy weight TOPSIS method is adopted to calculate the high-quality development level of 113 resource-based prefecture-level cities in China from 2003–2018. The high-quality development level of resource-based cities is evaluated from the perspectives of trends over time, region, and city type; further, the high-quality development level of all cities is rated [1]. The spatial patterns of high-quality development level of resource-based cities are analyzed using ArcGIS 10.2 software to examine the trend of hierarchical spatial distribution (Fig 2).

The quality level shows an upward trend over time, this is consistent. In 2003, the high-quality development level of resource-based cities ranged from 0.235–0.395. There are 97 low-level cities with high-quality development and 16 low-level cities, mainly in the northeastern and central regions. Compared with 2003, the high-quality development level of resource-based cities improved by 2008 but remained low, with scores ranging from 0.306–0.504; moreover, seven low-level cities remained, mainly in the southern region. The number of cities with low-level quality development increases to 105, accounting for 93% of all cities, and Xinyu City has a higher level of high-quality development. Compared to 2008, the high-quality development level of resource-based cities in 2013 is generally higher, with scores between 0.483 and 0.667, and the number of cities with lower levels of high-quality development reduce to two—Ya'an and Baise Cities. The number of cities with higher levels of development increases to 105, accounting for 93% of all the cities; the number of cities with high-quality development and high levels increases to six, mainly in the northeastern and eastern regions. Compared with 2013, the high-quality development level of resource-based cities is further improved in 2018, with a score of 0.522–0.741. Although the number of cities with higher levels of high-quality development decrease to 89, accounting for 79% of all cities, the high-level cities increase to 24, mainly in Gansu, Sichuan, Anhui, Inner Mongolia, Guizhou, and other provinces.

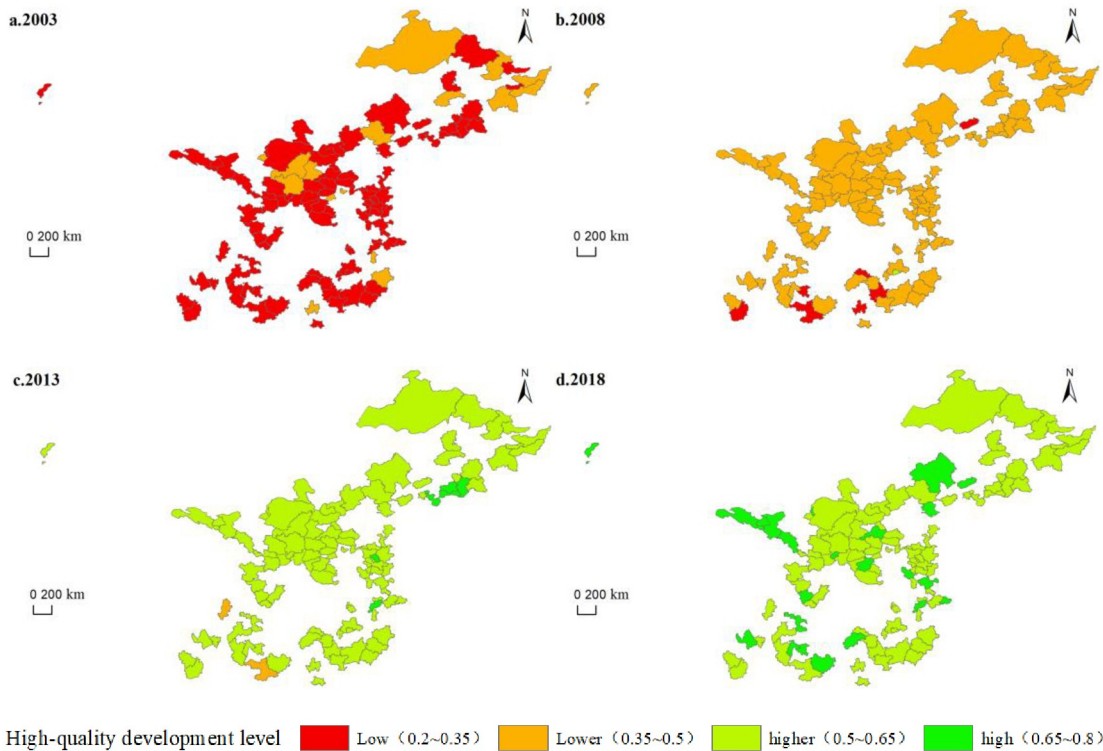

**Fig 2. Evolution trend of spatial distribution of high-quality development levels in resource-based cities.**

The high-quality development level of each city during the study period generally shows an upward trend, and the rate of improvement shows the spatial trend in the order of northwest > southwest > east > central > northeast. From 2003–2018, the high-quality development level of each province improves to varying degrees. Among them, the Guizhou Province has the largest increase, with an average annual growth rate of 9.38%, while the Heilongjiang Province has the smallest increase at 4%. Ten provinces have high-quality development levels that increased by more than 100%. Luzhou City had the largest increase in the high-quality development score, with 0.235 in 2003 and 0.683 in 2018, and an average annual growth rate of 11.94%. The smallest increase from 0.382 in 2003 to 0.529 in 2018 is observed in Shuangyashan, with an average annual growth rate of 2.38%. The research findings indicate that the overall level of high-quality development in China's resource-based cities has shown an upward trend, but there is still a large gap between different regions.

The high-quality development levels of the four types of resource-based cities increase during the study period, and the trend of the average annual growth rate of high-quality development levels is in the order of mature > regeneration > growth > decline city. The average high-quality development level of mature cities increases from 0.308 in 2003 to 0.625 in 2018, with an average annual growth rate of 6.43%. Their economic development is better, with a high level of high-quality development. The intermediate high-quality development level of regenerative cities increases from 0.311 in 2003 to 0.63 in 2018, with an average annual growth rate of 6.4%. High-quality development is achieved by changing the economic development mode. The average annual growth rate of the high-quality development level of growing and declining cities is low at 5.67% and 5.47%, respectively. Growth cities are in a phase of rising resource development and lack economic development momentum. The declining cities are in the exploratory stage of economic transformation and development, with relatively low high-quality development.

**4.1.2 Spatial correlation test of high-quality development of resource-based cities.**
Global spatial autocorrelation. This study uses ArcGIS10.2 and Geoda1.14 to calculate the adjacency weight matrix. However, because some resource-based cities are not suitable for the spatial adjacency matrix, the cities that appeared as "isolated islands" on the map (Hulunbuir, Fuxin, Huludao, Hebi, Hezhou, Yunfu, Ya'an, and Karamay Cities) establish a connection with the nearest resource-based city, and this further modifies the adjacency weight matrix. The Moran's I index of the high-quality development level of resource-based cities from 2003–2018 is calculated in stata15.0. Moran's I index of the high-quality development level of the 113 resource-based prefecture-level cities in the country is positive, with a fluctuating trend of first rising and then decreasing, and the spatial correlation also shows an initially increasing and then decreasing trend (Table 3). Results for 10 years are significant, indicating that the high-quality development level of resource-based cities during the study period is spatially correlated.

Local spatial autocorrelation. Stata 15.0 software is used to draw Moran's I scatterplot of the high-quality development level of 113 resource-based prefecture-level cities in China, and Geoda1.14 is used to draw the LISA cluster map in local Moran's I to analyze its spatial agglomeration characteristics. For 2003, 2008, 2013, and 2018 (Figs 3 and 4), most resource-based cities are in the first and third quadrants, corresponding to the high-high and low-low aggregation states. Therefore, the overall spatial agglomeration phenomenon is positive, which is consistent with the test results of global autocorrelation.

The spatial relationship between the high-quality development level of resource-based cities from 2003–2018 is positive. In 2003, the high-quality development level of resource-based cities is mainly concentrated in the first quadrant, with 37 cities in the high-high aggregation state, and 4 cities—Heihe, Qitaihe, Jixi, and Ordos Cities—with significant agglomeration,

**Table 3. Moran's I index of high-quality development level of resource-based cities from 2003–2018.**

| Year | I | z | p-value* |
|------|------|------|---------|
| 2003 | 0.164 | 1.870 | 0.031 |
| 2004 | 0.077 | 0.932 | 0.176 |
| 2005 | 0.161 | 1.844 | 0.033 |
| 2006 | 0.100 | 1.186 | 0.118 |
| 2007 | 0.262 | 2.940 | 0.002 |
| 2008 | 0.397 | 4.406 | 0.000 |
| 2009 | 0.318 | 3.560 | 0.000 |
| 2010 | 0.327 | 3.657 | 0.000 |
| 2011 | 0.255 | 2.862 | 0.002 |
| 2012 | 0.328 | 3.644 | 0.000 |
| 2013 | 0.140 | 1.610 | 0.054 |
| 2014 | 0.305 | 3.403 | 0.000 |
| 2015 | 0.076 | 0.915 | 0.180 |
| 2016 | 0.001 | 0.107 | 0.457 |
| 2017 | 0.055 | 0.689 | 0.246 |
| 2018 | 0.287 | 3.216 | 0.001 |

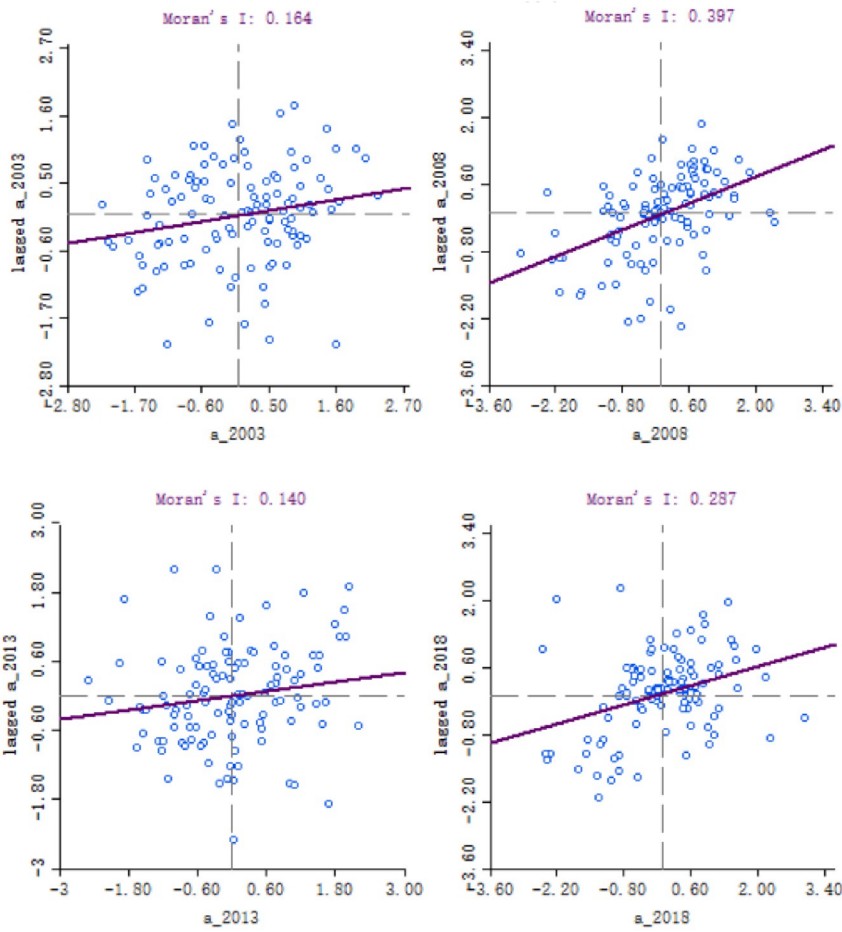

**Fig 3. Moran scatter plot of high-quality development level of resource-based prefecture- level cities from 2003–2018.**

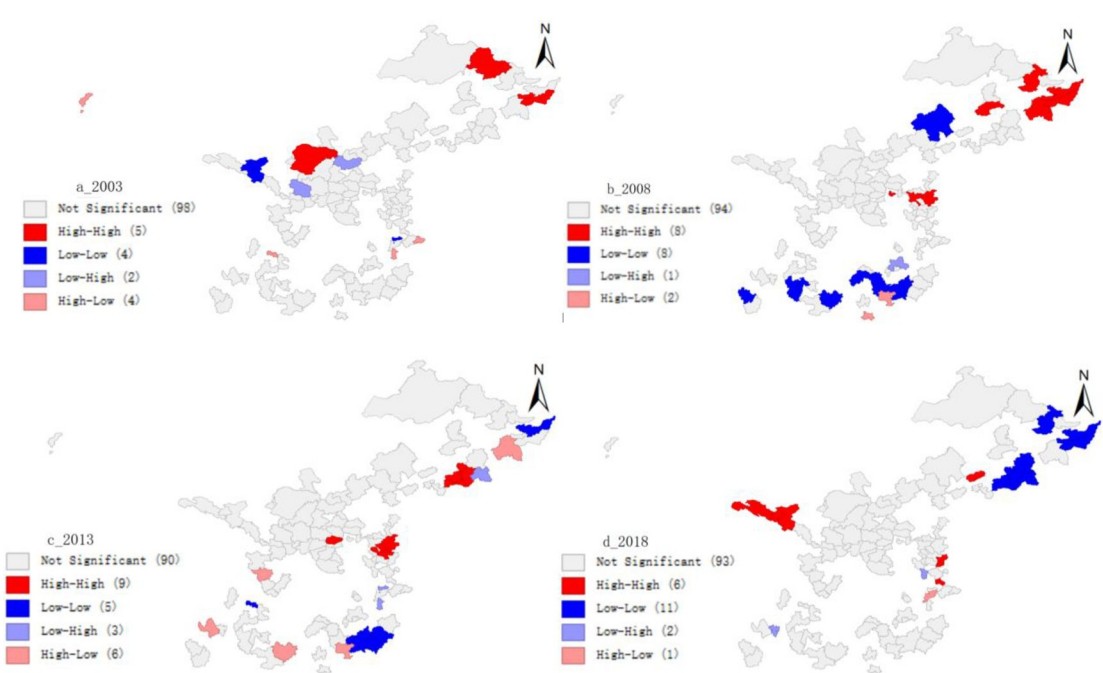

**Fig 4. LISA cluster diagram of high-quality development level of resource-based prefecture-level cities from 2003–2018.**

mainly distributed in the northeastern region; 24 cities in the second quadrant with low-high agglomeration, and 2 cities—Xinzhou and Qingyang Cities—with significant agglomeration; 28 cities in the third quadrant with low-low agglomeration, and 3 cities—Jinchang, Wuwei, and Tongling Cities—with significant agglomeration; and 24 cities in the fourth quadrant with high-low agglomeration, and 4 cities—Karamay, Zigong, Jingdezhen, and Huzhou Cities— with significant accumulations.

## 4.2 Empirical test of factors influencing the high-quality development of resource-based cities

**4.2.1 Model suitability test.** To test for multicollinearity between variables, this study measures the variance inflation factor (VIF) before running the econometric model. The largest of the seven variables is 1.46, which is much smaller than 10, indicating that multicollinearity is not a concern. Before the spatial econometric analysis of the factors affecting the high-quality development level of resource-based cities, a spatial econometric model must be selected. This study uses Stata15.0 to select the analysis model (Table 4). Specifically, the LM test results reveal that the Spatial Dubin Model should be used, and the Hausman test results are rejected. Therefore, the Fixed Effect Model is selected and the LR test results are all significant, indicating that the Spatial Dubin Model would not degenerate; from the Wald test, the Spatial Dubin Model simplification is rejected. Therefore, the Fixed Effects Spatial Dubin Model is used to analyze the factors influencing the high-quality development level of resource-based cities.

**4.2.2 Analysis of regression results of spatial econometric model for high-quality development of resource-based cities.** The larger the L-likelihood value, the better the fitting effect; the Space-time Double Fixed Effect Model (2357.3649) is better than the Time- (2085.1141) and Space- (2039.0692) Fixed Effect Models. Therefore, the Space-time Double

Table 4. Correlation test results of panel data model.

| Year | I | z | p-value* |
|------|------|------|---------|
| 2003 | 0.164 | 1.870 | 0.031 |
| 2004 | 0.077 | 0.932 | 0.176 |
| 2005 | 0.161 | 1.844 | 0.033 |
| 2006 | 0.100 | 1.186 | 0.118 |
| 2007 | 0.262 | 2.940 | 0.002 |
| 2008 | 0.397 | 4.406 | 0.000 |
| 2009 | 0.318 | 3.560 | 0.000 |
| 2010 | 0.327 | 3.657 | 0.000 |
| 2011 | 0.255 | 2.862 | 0.002 |
| 2012 | 0.328 | 3.644 | 0.000 |
| 2013 | 0.140 | 1.610 | 0.054 |
| 2014 | 0.305 | 3.403 | 0.000 |
| 2015 | 0.076 | 0.915 | 0.180 |
| 2016 | 0.001 | 0.107 | 0.457 |
| 2017 | 0.055 | 0.689 | 0.246 |
| 2018 | 0.287 | 3.216 | 0.001 |

Note:

\***P < 0.01,

\**P < 0.05,

\*P < 0.1;

the values in brackets are the corresponding z-statistics.

Fixed Spatial Dubin Model is used to analyze the influencing factors of the high-quality development level of resource-based cities (Table 5).

The regression coefficient of industrial structure level (IS) is positive, but insignificant, suggesting a possible increase in the proportion of the secondary industry's added value in the GDP, but the promotion of the high-quality development level of resource-based cities is insignificant. Resource-based cities have an excellent industrial foundation, and the GDP is mainly composed of the secondary industry. The greater the proportion of the added value of the secondary industry in GDP, the higher the level of economic development. However, additionally, there are many heavy industrial enterprises in the secondary industry of resource-based cities, which cause significant energy consumption and pollution, and are not conducive to green development in high-quality development.

The regression coefficient of the scientific and technological innovation level (RD) is 0.019, which passed the 1% significance level test. Increased investment in scientific and technological research and development, and continuously improving the level of innovation, can strengthen technological transformation and application and the ability to significantly improve the level of high-quality development. The research findings indicate that technological progress plays a positive role in promoting high-quality development.

The human capital level (HR) regression coefficient is 0.023, which passed the significance level test of 1%. An increase in regional education can strengthen the innovation drive and promote the high-quality development of cities.

The regression coefficient of the urbanization level (UPN) is 0.209 and passed the 1% significance level test. Therefore, with urbanization, the flow of labor, capital, technology, and other

**Table 5. Regression results of the Spatial Doberman Model.**

| Variable | Main | Wx | Spatial | Variance |
|---|---|---|---|---|
| ln (IS) | 0.018 (0.28) | -0.044**(0.04) | | |
| ln (RD) | 0.019***(0.00) | 0.003 (0.48) | | |
| ln (HR) | 0.023***(0.00) | 0.001 (0.88) | | |
| ln (UPN) | 0.209*** (0.00) | 0.102*** (0.00) | | |
| ln (ER) | -0.004*(0.07) | -0.009***(0.01) | | |
| ln (IEP) | 0.009*** (0.00) | -0.007*(0.06) | | |
| ln (IT) | -0.006 (0.52) | -0.005 (0.68) | | |
| rho | | | 0.076*** (0.00) | |
| sigma2_e | | | | 0.004*** (0.00) |
| Observations | 1,808 | 1,808 | 1,808 | 1,808 |
| R-squared | 0.320 | 0.320 | 0.320 | 0.320 |
| Number of CODE | 113 | 113 | 113 | 113 |

Note:

\*\*\*P < 0.01,

\*\*P < 0.05,

\*P < 0.1;

the values in brackets are the corresponding z-statistics.

elements and resources between urban and rural areas is promoted and the gap between urban and rural areas is narrowed, and the level of high-quality development improved.

The regression coefficient of environmental regulation (ER) is less than 0 and significant at the 10% level. Environmental regulation is beneficial to the improvement of the ecological environment and green development to a certain extent. However, a lack of strict local environmental protection law enforcement and "one-size-fits-all" measures for environmental protection partially offsets the promotion of environmental pollution control investment to the level of high-quality development and even inhibits high-quality development.

The regression coefficient of openness to the outside world (IEP) is 0.009, which passed the 1% significance level, indicating that the higher the openness, the smoother the flow of capital and the more foreign capital injected into high-quality development. This promotes the high-quality development of resource-based cities.

The regression coefficient of infrastructure level (IT) is negative and not significant. Increases in highway traffic density did not have an apparent inhibitory effect on the high-quality development level of resource-based cities. This can be attributed to investment in infrastructure, such as roads, crowding out other investments, resulting in a decline in resource allocation efficiency and inhibition of the quality of economic growth of resource-based cities.

**4.2.3 Robustness test.** To avoid the problem of reverse causality, following the approach of Zhu and Jing [51], this study selects the highest second-order and third-order spatial lag terms as instrumental variables and employs the generalized spatial two-stage least squares method to conduct an endogeneity test. The results of the robustness test regression are shown in Table 6, and the spatial lag term of high-quality development is significant at the 1% level. The regression results of each influencing factor are basically consistent with the baseline regression results, indicating that there is no reverse causality between the variables affecting high-quality development and high-quality development itself, and the baseline regression results are robust.

**Table 6. Robustness test results.**

| Variable | Second-order Lag | Third-order Lag |
|---|---|---|
| W*y | 0.082*** (0.00) | 0.081*** (0.00) |
| ln (IS) | 0.099 (0.53) | 0.098 (0.51) |
| ln (RD) | 0.073*** (0.00) | 0.0731*** (0.00) |
| ln (HR) | 0.088*** (0.00) | 0.0876*** (0.00) |
| ln (UPN) | 0.592*** (0.00) | 0.592*** (0.00) |
| ln (ER) | -0.006 (0.107) | -0.006*** (0.104) |
| ln (IEP) | 0.010*** (0.00) | 0.010*** (0.00) |
| ln (IT) | -0.013 (0.75) | -0.013 (0.74) |
| Observations | 1808 | 1808 |
| R-squared | 0.108 | 0.108 |
| Number of CODE | 113 | 113 |

Note:

***$P < 0.01$,

**$P < 0.05$,

*$P < 0.1$;

the values in brackets are the corresponding z-statistics.

**4.2.4 Spatial spillover effect analysis of factors influencing the high-quality development of resource-based cities.** Pace and LeSage [52] believe that the variable relationship of the Spatial Dubin Model should be described in terms of direct, indirect, and total effects. The direct effect refers to how changes in the independent variable of a certain place will affect the dependent variable of that place. The indirect effect refers to how changes in independent variables of a certain place affect the dependent variables of adjacent areas. The spatial effect decomposition estimation results of the Spatial Dubin Model are shown in Table 7.

**Table 7. Estimation results of direct and indirect effects of the Spatial Dubin Model.**

| Variable | Direct effect | Indirect effect | Total effect |
|---|---|---|---|
| ln (IS) | 0.017(0.31) | -0.046**(0.04) | -0.029(0.23) |
| ln (RD) | 0.019***(0.00) | 0.005(0.24) | 0.024***(0.00) |
| ln (HR) | 0.023***(0.00) | 0.004(0.68) | 0.027**(0.01) |
| ln (UPN) | 0.213***(0.00) | 0.122***(0.00) | 0.335***(0.00) |
| ln (ER) | -0.005*(0.05) | -0.009***(0.01) | -0.014***(0.00) |
| ln (IEP) | 0.009***(0.00) | -0.007*(0.07) | 0.002(0.61) |
| ln (IT) | -0.006(0.53) | -0.005(0.66) | -0.011(0.39) |
| Observations | 1,808 | 1,808 | 1,808 |
| R-squared | 0.320 | 0.320 | 0.320 |
| Number of CODE | 113 | 113 | 113 |

Note:

***$P < 0.01$,

**$P < 0.05$,

*$P < 0.1$;

the values in brackets are the corresponding z-statistics.

The regression coefficient of the direct effect of industrial structure level (IS) is positive, failing the significance test. The indirect effect is negative, passing the 1% significance test. The industrial structure level has a significant negative spillover effect on the high-quality development level of adjacent resource-based cities. Due to the large demand for labor in the secondary industry, increases in the output value of secondary industries can drive local economic development and absorb the labor resources in the surrounding areas, thereby inhibiting the high-quality development of adjacent areas.

The regression coefficients of the direct and indirect effects of the scientific and technological innovation level (RD) are positive, but the indirect effect is not significant. As resource-based cities expand their investment in scientific research funds and promote local scientific and technological research and development, scientific research institutions and achievements are not effectively shared, and therefore the promotion effect on the high-quality development level of adjacent areas is not obvious.

The regression coefficients of the direct and indirect effects of human capital level (HR) are positive, but the indirect effect is not significant. Improving the human capital level is an effective way to improve labor productivity and input-output levels. During the economic transformation of resource-based cities, improvements to labor productivity can significantly promote local high-quality development. However, it has no significant impact on the high-quality development of the adjacent areas.

The level of urbanization (UPN) has a significant positive effect on the high-quality development of both local and neighboring resource-based cities. With the improvement of the local urbanization level, human resources and factor resources gradually increase, and, to a certain extent, can spill over to surrounding areas. This can narrow the income gap between local urban and rural areas, improve the level of high-quality development, and promote high-quality development in neighboring areas.

Environmental regulation (ER)has a significant negative inhibitory effect on the high-quality development of local and adjacent resource-based cities. With improved local environmental regulations, pollution dumping may still occur, which is not conducive to the green development of adjacent areas, causing a negative spillover effect.

The level of openness to the outside world (IEP) plays a significant positive role in promoting the local, high-quality development level and has a significant negative spillover effect on adjacent resource-based cities. An increased openness to the outside world increases the local introduction of advanced science and technology. This agglomeration effect restrains the high-quality development of the surrounding areas.

The regression coefficients of the direct and indirect effects of infrastructure level (IT) are negative and insignificant. Increases in local highway traffic density affect the efficiency of resource allocation. However, this will not have a pronounced effect on the high-quality development level of adjacent resource-based cities.

**4.2.5 Regression analysis of influencing factors by city type.** A spatial distance matrix is selected as the spatial weight matrix. In addition to the spatial lag model (SAR) for growing cities, the spatial error model (SEM) is selected for mature, declining, and regenerative cities. The econometric test results of the influencing factors show that the effects of various factors on the high-quality development level of different resource-based cities have similarities and apparent differences (Table 8).

The positive factors affecting the high-quality development of the four resource-based cities were generally at the level of industrial structure, human capital, urbanization, and environmental regulation. Except for the environmental law, the others are consistent with the overall results. The role of environmental law in the high-quality development of mature cities passed the significance test. An increase in government investment in ecological governance is

**Table 8. Regression results of a spatial econometric model of high-quality development level of various types of cities.**

| Variable | Growth type | Mature type | Recessionary type | Regenerative type |
|---|---|---|---|---|
| ln (IS) | 0.043*(0.07) | -0.005(0.62) | 0.104***(0.00) | 0.079*(0.07) |
| ln (RD) | -0.009(0.22) | 0.005(0.22) | -0.006(0.34) | 0.013*(0.07) |
| ln (HR) | 0.004(0.53) | 0.008**(0.04) | -0.009*(0.07) | 0.011*(0.08) |
| ln (UPN) | 0.099***(0.00) | 0.055***(0.00) | 0.055***(0.01) | 0.079***(0.01) |
| ln (ER) | 0.002(0.62) | 0.006***(0.01) | 0.003(0.41) | -0.001(0.85) |
| ln (IEP) | -0.007**(0.04) | -0.007***(0.00) | 0.008***(0.00) | -0.003(0.69) |
| ln (IT) | -0.017**(0.05) | 0.003(0.44) | -0.024***(0.01) | -0.006(0.55) |
| Spatial rho | -0.175**(0.04) | -0.597***(0.00) | 0.114*(0.09) | -0.290**(0.01) |
| Variance sigma2_e | 0.006***(0.00) | 0.006***(0.00) | 0.006***(0.00) | 0.004***(0.00) |
| Observations | 224 | 976 | 368 | 240 |
| R-squared | 0.164 | 0.346 | 0.070 | 0.204 |
| Number of CODE | 14 | 61 | 23 | 15 |

Note:

***$P < 0.01$,

**$P < 0.05$,

*$P < 0.1$;

the values in brackets are the corresponding z-statistics.

conducive to the technological progress of energy conservation, emission reduction, and green development of resource-based industries in mature cities. This can significantly promote high-quality development.

In general terms, the negative factors affecting the high-quality development of the four resource-based cities are the level of openness to the outside world and infrastructure. The infrastructure level is consistent with the overall test results. Openness to the outside world significantly inhibits the high-quality development of growing and mature cities. This is because urban development depends on the advantages of traditional resources. When the proportion of total imports and exports in GDP increases, the low added value of most export products is not conducive to the optimization of industrial structure and hinders the improvement of high-quality development.

## 5. Conclusion and discussion

### 5.1 Conclusion

This study uses panel data for 113 resource-based prefecture-level cities in China from 2003–2018 and the entropy weight TOPSIS method and spatial correlation analysis to explore the spatiotemporal evolution characteristics of high-quality development and a spatial econometric model to analyze the impact of its high influence factors of quality development. We can able to draw the following conclusions:

1. During the study period, the overall high-quality development level of resource-based cities in China show an upward trend, with the order of the spatial pattern being of Northwest > Southwest > East > Middle > Northeast; the average annual growth rate of the high-quality development level of the four types of resource-based cities is in the order of maturity > regeneration > growth > decline.

2. The results of the spatial correlation test show that the high-quality development of resource-based cities generally shows a positive spatial correlation, and Moran's I index shows a fluctuating trend. In the local spatial autocorrelation test, most resource-based cities are in a state of high aggregation and low oligomerization, indicating that the spatial spillover effect of high-quality development is noticeable, especially for cities with significant high-high aggregation.

3. The results of influencing factor analysis show that the level of scientific and technological innovation, human capital, urbanization, and openness to the outside world plays a significant role in promoting the level of high-quality development; environmental regulation played a significant inhibitory role, while the industrial structure and infrastructure level are not significant. The level of industrial structure, environmental regulation, and openness to the outside world have a significant negative spillover effect, and the level of urbanization has a significant positive spillover effect. All factors have some common effect on the high-quality development of different types of resource-based cities.

## 5.2 Discussion

Based on the five major development concepts, we examine the high-quality development of resource-based cities across China by constructing a complete high-quality development evaluation index system and systematically analyze the spatio-temporal evolution characteristics of high-quality development levels and the impact mechanisms of different types of resource-based cities.

The study reveals that resource-based cities exhibit an upward trend in the level of high-quality development. Significant variations in high-quality development are observed among different regions, consistent with findings by Su [53] and Cui et al. [54]. Furthermore, spatial correlation analysis indicates an overall positive spatial correlation in the high-quality development of resource-based cities, confirming results from Ding and Liu [55]. Notably, there is a distinct spatial spillover effect in high-quality development. Exploring the influencing mechanisms of high-quality development in resource-based cities, it is found that levels of technological innovation, human capital, urbanization, and openness significantly promote high-quality development [56]. Conversely, environmental regulation exerts a notable inhibitory effect, contrary to the results of Li et al. [57]. This discrepancy may be attributed to the threshold effect of environmental regulation, leading to variations in conclusions at different study periods.

The differences between this study and the existing literature also point out the direction for future research. In the future, further exploration will be conducted in two aspects: first, the reasons for the differences in the impact mechanisms studied in this paper and those of other scholars should be analyzed, and it may be related to the existence of threshold effects. Therefore, future research should focus on whether there are threshold effects in the impact mechanism. In addition, facing various factors such as the proposal of China's "dual carbon" goal and the construction of the "14th Five-Year Plan", there is still a large space for promoting and improving the high-quality development of resource-based cities. The next step of the research should take these factors into consideration and solve current and future practical problems.

## 6. Policy implications

Based on our results, we propose the following policy recommendations: (1) Drive innovation and upgrade industries. Strengthen innovation as a driver for upgrading traditional resource-

based industries with green technology. Enhance the technology innovation system, focus on human capital investment, and cultivate independent innovation. (2) Promote interregional resource flow. Improve interregional coordination for efficient resource circulation between urban and rural areas, fostering conditions for industrial upgrades. (3) Enhance pollution control and green tech. Strengthen coordinated pollution control by improving energy efficiency and environmental practices, minimizing spillover effects between cities, and encouraging the use of green technologies. (4) Boost openness and investment. Elevate openness levels through strategic investment attraction in resource-saving and environmentally friendly enterprises, promoting energy-saving technologies and cross-border cooperation. (5) Establish a shared development system. Build a shared development system through new infrastructure like "5G" to enhance public services, education, healthcare, and infrastructure investment efficiency. (6) Optimize human capital and transformation. Enhance human capital in resource-based cities through higher education, supporting transformation towards resource-based and innovative industries, and optimizing talent resource allocation.

## Supporting information

**S1 Data.**
(XLSX)

## Author Contributions

**Data curation:** Changchun Lin.

**Funding acquisition:** Xiuli Liu.

**Project administration:** Xiuli Liu.

**Software:** Xin Wang.

**Writing – original draft:** Jun Liu.

**Writing – review & editing:** Jun Liu.

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
