## [Decision Letter · Decision Letter 0]

20 Mar 2023

PONE-D-23-05547Spatiotemporal differentiation and analysis of factors influencing high-quality development of resource-based cities: An empirical study based in ChinaPLOS ONE

Dear Dr. liu,

Thank you for submitting your manuscript to PLOS ONE. After careful consideration, we feel that it has merit but does not fully meet PLOS ONE’s publication criteria as it currently stands. Therefore, we invite you to submit a revised version of the manuscript that addresses the points raised during the review process.

We look forward to receiving your revised manuscript.

Kind regards,

Jing Cheng

Academic Editor

PLOS ONE

Journal Requirements:

"The authors received support from the National Natural Science Foundation of China (Grant Nos. 42001257 and 71874119), Philosophy and Social Sciences Research of Higher Learning Institutions of Shanxi (Grant No. 20210115), and the Shanxi Postgraduate Education Innovation Project (Grant No. 2021Y571). We also thank our anonymous reviewers and the editor for their helpful suggestions"

"The authors received support from the National Natural Science Foundation of China (Grant Nos. 42001257 and 71874119), Philosophy and Social Sciences Research of Higher Learning Institutions of Shanxi (Grant No. 20210115), and the Shanxi Postgraduate Education Innovation Project (Grant No. 2021Y571). We also thank our anonymous reviewers and the editor for their helpful suggestions."

Reviewers' comments:

Reviewer's Responses to Questions

**Comments to the Author**

1. Is the manuscript technically sound, and do the data support the conclusions?

Reviewer #1: Yes

Reviewer #2: Partly

2. Has the statistical analysis been performed appropriately and rigorously? 

Reviewer #1: Yes

Reviewer #2: N/A

3. Have the authors made all data underlying the findings in their manuscript fully available?

Reviewer #1: Yes

Reviewer #2: No

4. Is the manuscript presented in an intelligible fashion and written in standard English?

Reviewer #1: Yes

Reviewer #2: Yes

5. Review Comments to the Author

Reviewer #1: This paper aimed to examine resource-based prefecture-level cities across China to systematically analyze the spatiotemporal evolutionary characteristics of high-qualitydevelopment levels for different types of resource-based cities using data were collected from 2003–2018 for 113 resource-based prefecture-level cities from across

China.I have some comments.

1.Why the data was chosen just before 2018.

2.The definition of a resource city started in 2007, so why don't the data start in 2007.

3.five development concepts, refer to ?Please define such Chinese expressions.

4.Separate the introduction from the literature review.

5.Research gaps and research implications should be given thoroughly.

6.The results of the appropriate discussion are studied by comparison.

Reviewer #2: (1) The logic of the literature review is not clear enough. The author should pay attention to the following questions: ① Is high-quality development equivalent to sustainable development? If there are differences between the two, the key words should be clearly defined in the process of literature review and discussion, and should not be arbitrarily replaced If the purpose of literature review is to clarify the factors that affect the high-quality development of resource-based cities, it should focus on sorting out the content rather than focusing on empirical methods.

(2) On the construction of the indicator system for high-quality development, the author has expanded the generalized high-quality indicator development system on the basis of the traditional high-quality development indicator system. However, the author has not clearly expressed the reasons and basis for the expansion of the article, and should be discussed in advance in the content of the literature review to make it reasonable to rely on.

(3) For the selection of data years, the data from 2003 to 2018 are somewhat outdated. Although the author retains the integrity of the study area, it also greatly sacrifices the forward-looking nature of the article.

(4) The author does not consider endogenous issues. Is there a reverse causal relationship between variables that affect high-quality development and high-quality development? Failure to consider endogenous issues is likely to cause significant errors in the coefficients of the article.

6. PLOS authors have the option to publish the peer review history of their article (what does this mean?). If published, this will include your full peer review and any attached files.

Reviewer #1: No

Reviewer #2: No

---

## [Author Response · Author response to Decision Letter 0]

20 Jul 2023

Dear Editor,

We are grateful to you and the other reviewers for your critical comments and thoughtful suggestions on our manuscript. We have carefully revised the manuscript based on your feedbacks. The revised portions of the manuscript are shown in red. We hope the revised manuscript satisfies the standards of the journal.

Below, we provide point-by-point responses to reviewer#1and reviewer#2’s comments and questions. We hope that these revisions are satisfactory and the revised manuscript could meet the publishing standards of the PLOS one.

Thank you very much for your assistance with our paper.

Responses to Reviewer #1:

1.Why the data was chosen just before 2018.

Response: According to the reviewer's comments, an explanation was provided for the selection of data prior to 2018. The reason is as follows: due to the outbreak of COVID-19 in China in 2019, the data for 2019-2020 is relatively distorted, and some data for 2021-2022 has not yet been released, resulting in the final selection of data prior to 2018.

2.The definition of a resource city started in 2007, so why don't the data start in 2007.

Response: According to the reviewer's comments, an explanation was provided for the selection of data prior to 2018. The reasons are as follows: First, although the definition of resource-based cities began in 2007, China's attention to resource-based cities dates back earlier than 2007. In 2001, Fuxin was designated as the first resource-depleted city in China's national economic transformation pilot cities. Since then, China has begun to pay attention to the transformation and development of resource-based cities. Secondly, the reason why 2001 was not chosen as the starting year is that China's socialist market economy system was basically established after 2002, and the domestic institutional environment in China was relatively stable. In addition, at the end of 2001, China joined the World Trade Organization, so the external environment faced by China during the sample period was basically the same. Therefore, selecting 2003 as the starting year of the study is appropriate.

3.five development concepts, refer to ?Please define such Chinese expressions.

Response: According to the reviewer's comments, a thorough review was conducted on the connotations of the Five Development Concepts. The revisions are as follows:

The Five Development Concepts refer to the five development strategies proposed by the Communist Party of China in the context of the new era, namely innovation, coordination, green development, openness, and sharing. These concepts aim to guide the development of China's economy, politics, culture, and other fields, and promote China's transition to high-quality development [38]. Specifically, innovation refers to promoting technological and industrial innovation to improve the quality and efficiency of economic development; coordination refers to promoting coordinated development between urban and rural areas, industries, regions, and sectors; green development refers to promoting ecological civilization construction, protecting the environment, and achieving coordinated development between economic development and environmental protection; openness refers to deepening opening-up, actively participating in global economic governance, and promoting economic globalization; sharing refers to promoting social fairness and justice, strengthening social security, and achieving more equitable and inclusive distribution of development outcomes. The Five Development Concepts are interconnected and mutually reinforcing, forming an integrated whole with inherent connections [39].

4.Separate the introduction from the literature review.

Response: According to the reviewer's comments, the introduction should be separated from the literature review. The revised version is as follows:

Currently, research in academia on high-quality development mainly focuses on its connotation, meaning, and evaluation [7-9], and a unified evaluation index system for high-quality development has not yet been formed, especially for resource-based regions. There is a lack of research on the evaluation index system for high-quality development in resource-based regions. In addition, scholars often use the national, provincial, and urban agglomeration levels as the research areas, analyzing the transformation and development of resource-based cities from the perspectives of labor quality [10], environmental pollution [11, 12], and education quality [13], but relatively little research has been done on the factors that affect high-quality development. Based on this, this paper takes 113 resource-based prefecture-level cities in China from 2003 to 2018 as samples, and uses entropy weight-TOPSIS method and spatial correlation analysis to construct a high-quality development evaluation system, and studies the spatio-temporal evolution trend of high-quality development in different types of cities. At the same time, a spatial econometric model is used to explore the role of industrial structure, technological innovation, human capital, urbanization, environmental regulation, openness to the outside world, and infrastructure in high-quality development. Finally, high-quality development suggestions are proposed based on empirical results.

The contributions of this study are: (1) Focusing on the "Five Major Development Concepts," a comprehensive and generalized evaluation index system for high-quality development of resource-based cities is constructed from multiple perspectives, which fills the gap in the current research on the evaluation index system for high-quality development of resource-based cities; (2) Including the vast majority of resource-based cities in China in the research sample makes the sample region more comprehensive and the study more practical; (3) Addressing the problem of a lack of research on the factors that influence high-quality development in resource-based cities and the inadequate selection of variables, this study selects the influencing factors as comprehensively, scientifically, and purposefully as possible, thereby analyzing the obstacles that resource-based cities face in the process of high-quality development and providing solutions.

The rest of this study is structured as follows. Part 2 introduces the relevant literature on high-quality development in resource-based cities. Part 3 describes the research methods, study areas, data sources, and variable selection. Part 4 presents the research structure and analysis, including spatio-temporal evolution analysis of high-quality development in resource-based cities and empirical testing of influencing factors. Part 5 is the conclusion and policy implications of the study.

Literature Review

The academic community's understanding of the connotation of high-quality development has undergone a transformation from focusing solely on the quality of economic growth to development quality. Initially, people's understanding of high-quality development was mostly focused on economic development, and they believed that when measuring regional development, using only per capita GDP was not comprehensive enough, and that people's lives and social welfare should also be included [14, 15]. As scholars have conducted more in-depth research on the connotation of high-quality development, they have gradually incorporated aspects such as ecological environment, society, and human capital into the evaluation index system for measuring regional high-quality development [16-18]. At this point, the academic community has reached a consensus on the definition of the connotation of high-quality development, generally believing that high-quality development is efficient and high-quality development that can meet people's material, social, and psychological needs. Currently, with the proposal of the five development concepts of innovation, coordination, green, openness, and sharing, the connotation of high-quality development has been further deepened [19]. Compared with the previous connotation of high-quality development, the five development concepts have more systematic and comprehensive characteristics, more emphasis on people-oriented development, and are more in line with the trend of the times, providing more comprehensive theoretical support for evaluating regional development. Regarding the evaluation of high-quality development, most scholars use evaluation methods such as entropy weight method [20-23] and TOPISI method [24, 25] to construct evaluation index systems and calculate the level of high-quality development, and a few scholars use methods such as DEA model [26] and coupling coordination degree analysis [27-29] to evaluate the development of regions with high quality. In addition, some scholars have conducted comparative studies on the relationship and differences between high-quality development and sustainable development, believing that early high-quality development was more focused on economic quality compared to sustainable development [30]. However, with the proposal of the five development concepts, the content and scope of high-quality development have continued to expand [31], leading to the gradual formation of a unified whole in which high-quality development and sustainable development are mutually coordinated and complementary.

The research on the mechanism of high-quality development of resource-based cities mainly focuses on institutional construction, financial support, and industrial structure. For example, Al-Ubaydli [32] and Wiens [33] found that to avoid the resource curse, it is possible to promote the transformation and development of resource-based cities through high-quality institutional design. Song Liying et al. [34] proposed to reduce the fiscal autonomy and expand the scale of special funds by studying the role of fiscal autonomy in total factor productivity for resource-based cities. Jin Bei [35] believes that the government plays an important role in high-quality development, especially in the western region, which relies heavily on government support. At the same time, he suggests that technological innovation is the vitality source of high-quality development. Yang et al. [36] found that the single industrial structure, inadequate attention to ecological protection, and low regulatory efficiency of mineral resource-based cities increase the transformation cost and are not conducive to high-quality development. In addition, regarding the development path of resource-based cities, Yang et al. [5] pointed out that the transformation and development of resource-based cities cannot be achieved without long-term financial and policy support from the government. By improving the high pollution problems of resource-based enterprises or forcing high-pollution enterprises to exit the market, resource-based cities can achieve a smooth transformation. Li et al. [37] proposed that resource-based cities should construct ecological environmental protection and ecological compensation mechanisms that consider the full life cycle of mineral resource development and utilization to coordinate sustainable development and ecological protection of resources.

5.Research gaps and research implications should be given thoroughly.

Response According to the reviewer's comments, research gaps and inspirations should be added to the conclusion and recommendation section. The revised version is as follows:

Based on our results, we propose the following policy recommendations: (1) Strengthen the driving force of innovation and cultivate new momentum for development. Strengthening the research and development of green technology [4] should help transform and upgrade traditional resource-based industries, and move the industrial structure from low-end to high-end [64]. Improve the technology innovation system and mechanism in resource-based cities and implement dynamic management, optimize the regional industrial structure. A focus on human capital investment to meet the development needs of strategic emerging industries can help cultivate the independent innovation ability of enterprises and realize the industrialization of high-tech [52]. (2) Strengthen interregional coordination and exchanges to achieve mutual promotion and expected progress. Promoting the effective circulation of resources between urban and rural areas can create good human capital conditions for upgrading industrial structures. (3) Strengthen the coordinated control of pollutants and promote green development. Improving the energy consumption of secondary industries and environmental pollution should control the negative spillover effect between cities, and strictly control the transfer of pollution to neighboring resource-based cities. Furthermore, the comprehensive use of desulfurization, low-carbon, energy-saving, and environmental protection technologies [65] can strengthen the coordinated control of environmental pollutants and improve the compensation mechanism for environmental governance. Additionally, effectively utilizing the advantages of resource extraction and deep processing technologies, combined with emerging industries, encouraging increased flow to the resource recycling industry, and high-end equipment manufacturing industry can aid in building a green, diversified, and complementary industrial system. (4) Coordinate the external and internal openings to improve the openness level. Strengthening investment attraction can help attract powerful foreign enterprises. Additionally, targeting investment in resource-saving and environment-friendly enterprises can bring further energy-saving, emission-reduction technologies, and advanced experience. Free flow between countries can strengthen cooperation in the fields of energy, financial industry, technological innovation, and emerging industries. (5) Build a shared development system to promote social equity. The construction of "5G" and other new infrastructure, with the help of "Internet +" thinking can improve the public service system, and strengthen education, medical care, culture, etc., and establish a sound shared development system. Further, infrastructure capital investment should be rationally allocated, to improve investment efficiency. (6) Improve the human capital level of resource-based cities, with a focus on optimizing its structure [55]. China's resource-based cities should rely on higher education to cultivate high-quality human capital, promote their transformation towards resource-based industries and development towards innovative industries, and promote the effective accumulation and optimal allocation of talent resources.

This paper provides a comprehensive examination and analysis of the high-quality development of resource-based cities based on existing data and theory. However, there are still limitations. In the future, further exploration will be conducted in two aspects: first, the reasons for the differences in the impact mechanisms studied in this paper and those of other scholars should be analyzed, and it may be related to the existence of threshold effects. Therefore, future research should focus on whether there are threshold effects in the impact mechanism. In addition, facing various factors such as the proposal of China's "dual carbon" goal and the construction of the "14th Five-Year Plan", there is still a large space for promoting and improving the high-quality development of resource-based cities. The next step of the research should take these factors into consideration and solve current and future practical problems.

6.The results of the appropriate discussion are studied by comparison.

Response According to the reviewer's comments, corresponding result discussions have been added to the results section of the article.

Dear Editor,

We are grateful to you and the other reviewers for your critical comments and thoughtful suggestions on our manuscript. We have carefully revised the manuscript based on your feedbacks. The revised portions of the manuscript are shown in red. We hope the revised manuscript satisfies the standards of the journal.

Below, we provide point-by-point responses to reviewer#2’s comments and questions. We hope that these revisions are satisfactory and the revised manuscript could meet the publish

---

## [Decision Letter · Decision Letter 1]

6 Sep 2023

PONE-D-23-05547R1Spatiotemporal differentiation and analysis of factors influencing high-quality development of resource-based cities: An empirical study based in ChinaPLOS ONE

Dear Dr. 刘,

Thank you for submitting your manuscript to PLOS ONE. After careful consideration, we feel that it has merit but does not fully meet PLOS ONE’s publication criteria as it currently stands. Therefore, we invite you to submit a revised version of the manuscript that addresses the points raised during the review process.

We look forward to receiving your revised manuscript.

Kind regards,

Jing Cheng

Academic Editor

PLOS ONE

Journal Requirements:

Reviewers' comments:

Reviewer's Responses to Questions

**Comments to the Author**

1. If the authors have adequately addressed your comments raised in a previous round of review and you feel that this manuscript is now acceptable for publication, you may indicate that here to bypass the “Comments to the Author” section, enter your conflict of interest statement in the “Confidential to Editor” section, and submit your "Accept" recommendation.

Reviewer #1: All comments have been addressed

Reviewer #3: (No Response)

2. Is the manuscript technically sound, and do the data support the conclusions?

Reviewer #1: Yes

Reviewer #3: Yes

3. Has the statistical analysis been performed appropriately and rigorously? 

Reviewer #1: Yes

Reviewer #3: Yes

4. Have the authors made all data underlying the findings in their manuscript fully available?

Reviewer #1: Yes

Reviewer #3: Yes

5. Is the manuscript presented in an intelligible fashion and written in standard English?

Reviewer #1: (No Response)

Reviewer #3: Yes

6. Review Comments to the Author

Reviewer #1: Resource-based cities often face challenges such as lack of resources and insufficient power for transformation and complex high-quality development. Further, research on the factors influencing high-quality development in such cities is limited. This paper aimed to examine resource-based prefecture-level cities across China to

systematically analyze the spatiotemporal evolutionary characteristics of high-quality development levels for different types of resource-based cities. To this end, data were collected from 2003–2018 for 113 resource-based prefecture-level cities from acrossChina, and the entropy weight TOPSIS method and spatial correlation analysis were used to conduct an in-depth analysis of the spatiotemporal evolution characteristics of high-quality development levels; further, we used spatial econometric models to empirically test the factors that influence high-quality development.The author has made sufficient revisions. I recommend acceptance.

Reviewer #3: The manuscript has been improved but it is recommended that authors work on the following suggestions:

1. The abstract is lengthy, I recommend that the author(s) focus on the salient points especially for the results.

2. In the introduction, line 75, kindly correct this: …, not only to solve their development problems but rather to …, not only to solve their developmental problems.

3. The objectives must be well stated.

4. The paragraph on the contribution of the study is not clear (line 95-106). Are the author(s) referring to the significance of the study?

5. The section on indicator system should be part of the literature while the specific indicators used and their ranges could be in the methods section.

6. Author(s) should check the version of ArcGIS used… it should be ArcGIS 10.2. This should be mentioned in the methods section and not the results. Stata is also mentioned in the results section but it would be more appropriate to mention the analytical software in the methods.

7. The version of ArcGIS used is not consistent. At a point, author(s) state it is 10.2 and in 10.3 at other points. Author(s) should kindly clarify this.

8. It is inappropriate to cite other scholarly works at the recommendations section. Author(s) should kindly revise this.

7. PLOS authors have the option to publish the peer review history of their article (what does this mean?). If published, this will include your full peer review and any attached files.

Reviewer #1: No

Reviewer #3: No

---

## [Author Response · Author response to Decision Letter 1]

3 Oct 2023

Response: Thank you for the comment and suggestion. We have rechecked the abstract part of the paper and re-organized it carefully.

Line 24-39:

Abstract：Resource-based cities often face challenges such as lack of resources and insufficient power for transformation and complex high-quality development. Further, research on the factors influencing high-quality development in such cities is limited. Thus, our approach utilized the entropy weight TOPSIS method, spatial correlation analysis, and spatial econometric models to explore key factors influencing high-quality development in China's resource-based cities. Key findings include: (1) An annual increase in overall high-quality development among Chinese resource-based cities, with the growth rate ranking as mature > regeneration > growth > declining cities. (2) Positive spatial correlation in high-quality development across resource-based cities, with fluctuating Moran's I index results indicating varying levels of clustering. (3) Scientific and technological innovation, human capital, urbanization, and openness to the global market significantly impact high-quality development. Distinct variations in how these factors influence different types of resource-based cities. This empirical research offers valuable insights for policymakers working to reduce regional disparities and promote high-quality development in resource-based cities.

Comment 2: In the introduction, line 75, kindly correct this: …, not only to solve their development problems but rather to …, not only to solve their developmental problems.

Response: We agree with the reviewer’s comments and suggestions. We have carefully revised the part.

Line 63-65:

Therefore, promoting the high-quality development of resource-based cities is necessary, not only to solve their developmental problems but rather to implement a national strategy.

Comment 3: The objectives must be well stated.

Response: We agree with the reviewer’s comments and suggestions. We have carefully revised the objectives.

Line 74-86:

To address these gaps, our objective is to examine resource-based prefecture-level cities across China to systematically analyze the spatiotemporal evolutionary characteristics of high-quality development levels and the influencing mechanism for different types of resource-based cities. This paper takes 113 resource-based prefecture-level cities in China from 2003 to 2018 as samples, and uses entropy weight-TOPSIS method and spatial correlation analysis to construct a high-quality development evaluation system, and studies the spatio-temporal evolution trend of high-quality development in different types of cities. At the same time, a spatial econometric model is used to explore the role of industrial structure, technological innovation, human capital, urbanization, environmental regulation, openness to the outside world, and infrastructure in high-quality development. Finally, high-quality development suggestions are proposed based on empirical results.

Comment 4: The paragraph on the contribution of the study is not clear (line 95-106). Are the author(s) referring to the significance of the study?

Response: Thanks to the reviewer's comments. We have reorganized the contributions to this paper. The examples are as follows:

Line 87-100:

The contributions of this study are: (1) We have developed an inclusive and widely applicable evaluation index system for assessing high-quality development in resource-based cities. This innovative system, rooted in the "Five Major Development Concepts," fills a critical void in the current research landscape, providing a holistic framework for evaluation. (2) By encompassing the majority of China's resource-based cities within our research sample, our study offers a comprehensive analysis that holds practical relevance for a broader spectrum of regions. This inclusivity enhances the applicability of our findings. (3) Our research addresses the critical knowledge gap concerning the factors that influence high-quality development in resource-based cities. We meticulously select these influencing factors, employing a scientific and purposive approach. Through this comprehensive analysis, we shed light on the challenges these cities face during their high-quality development journey and offer well-informed solutions. This contribution advances our understanding of resource-based city development.

Comment 5: The section on indicator system should be part of the literature while the specific indicators used and their ranges could be in the methods section.

Response：We appreciate your valuable comments and recommended references on our manuscript. After revisions, we placed the indicator system in the literature review section and the specific indicators as well as the scope in the methods section.

Line 122-144:

Currently, with the proposal of the five development concepts of innovation, coordination, green, openness, and sharing, the connotation of high-quality development has been further deepened [19]. The five development concepts refer to the five development strategies proposed by the Communist Party of China in the context of the new era, namely innovation, coordination, green development, openness, and sharing. These concepts aim to guide the development of China's economy, politics, culture, and other fields, and promote China's transition to high-quality development [38]. Specifically, innovation refers to promoting technological and industrial innovation to improve the quality and efficiency of economic development; coordination refers to promoting coordinated development between urban and rural areas, industries, regions, and sectors; green development refers to promoting ecological civilization construction, protecting the environment, and achieving coordinated development between economic development and environmental protection; openness refers to deepening opening-up, actively participating in global economic governance, and promoting economic globalization; sharing refers to promoting social fairness and justice, strengthening social security, and achieving more equitable and inclusive distribution of development outcomes. The Five Development Concepts are interconnected and mutually reinforcing, forming an integrated whole with inherent connections [20]. Compared with the previous connotation of high-quality development, the five development concepts have more systematic and comprehensive characteristics, more emphasis on people-oriented development, and are more in line with the trend of the times, providing more comprehensive theoretical support for evaluating regional development.

Table 1. Evaluation index system and weight of high-quality development of resource-based cities

Criterion layer Serial number Indicator layer Entropy Weight Method Weight Indicator properties

High-quality development based on "five development concepts" economy growth efficiency X1 GDP/social workers 0.025 +

 X2 GDP growth rate 0.037 +

 X3 Growth rate of social fixed asset investment 0.03 +

 growth level X4 Real GDP per capita 0.029 +

 X5 per capita fiscal revenue 0.034 +

 X6 gross industrial output per capita 0.028 +

 innovation innovative research and development X7 Science and technology expenditure 0.032 +

 X8 Number of granted patent applications 0.023 +

 X9 The number of green invention patents 0.029 +

 innovation potential X10 Education spending as a share of GDP 0.03 +

 X11 Number of teachers per 10,000 people 0.019 +

 transformation development X12 Energy consumption per ten thousand yuan of GDP 0.025 -

 X13 Employees in the tertiary industry among 10,000 people 0.045 +

 coordination regional coordination X14 Thiel Index 0.031 +

 demand coordination X15 Total per capita consumption 0.031 +

 X16 Engel's coefficient of urban residents 0.024 -

 X17 Year-end urban registered unemployment rate 0.032 -

 industrial structure X18 Primary and secondary industry coordination degree 0.027 -

 X19 Primary and tertiary industry coordination degree 0.027 -

 X20 Industrial development potential 0.025 +

 green resource consumption X21 Energy consumption per capita 0.021 -

 X22 Household water consumption per capita 0.017 +

 ecological environment X23 Total industrial "three wastes" emissions per 10,000 yuan of industrial added value / t 10,000 yuan -1 0.017 -

 X24 PM2.5 0.02 -

 X25 Green coverage rate of built-up area 0.025 +

 environmental governance X26 Comprehensive utilization rate of industrial solid waste 0.017 +

 X27 Centralized treatment rate of sewage treatment plant 0.022 +

 green living X28 Green area per capita 0.028 +

 X29 Harmless treatment rate of domestic pollution 0.023 +

 open trade open X30 Total import and export trade per capita 0.02 +

 investment open X31 Utilize the total amount of foreign investment/GDP 0.034 +

 X32 Number of foreign-invested enterprises 0.027 +

 shared public service X33 Per capita education expenditure of local finance 0.031 +

Comment 6: Author(s) should check the version of ArcGIS used… it should be ArcGIS 10.2. This should be mentioned in the methods section and not the results. Stata is also mentioned in the results section but it would be more appropriate to mention the analytical software in the methods.

Response：We appreciate the reviewer’s helpful and detailed comments. We check the version of ArcGIS and determine it to be ArcGIS 10.2, and make corresponding changes in the text. The use of Stata is mentioned in the methods section.

Line 373-375:

The spatial pattern changes of high-quality development level of resource-based cities were analyzed using ArcGIS 10.2 software to examine the trend of hierarchical spatial distribution (Fig. 2).

Line 432-433:

This study used ArcGIS10.2 and Geoda1.14 to calculate the adjacency weight matrix.

Line 336-337:

The data processing software used in this paper is Stata15.0.

Line 449-452:

Stata15.0 software was used to draw Moran's I scatterplot of the high-quality development level of 113 resource-based prefecture-level cities in China, and Geoda1.14 was used to draw the LISA cluster map in local Moran's I to analyze its spatial agglomeration characteristics.

Line 483:

This study used Stata15.0 to select the analysis model (Table 4).

Comment 7: The version of ArcGIS used is not consistent. At a point, author(s) state it is 10.2 and in 10.3 at other points. Author(s) should kindly clarify this.

Response：Thanks to the reviewer's comments, we used ArcGIS 10.2 software and we rechecked the article and revised it carefully.

Line 373-375:

The spatial pattern changes of high-quality development level of resource-based cities were analyzed using ArcGIS 10.2 software to examine the trend of hierarchical spatial distribution (Fig. 2).

Line 432-433:

This study used ArcGIS10.2 and Geoda1.14 to calculate the adjacency weight matrix.

Comment 8: It is inappropriate to cite other scholarly works at the recommendations section. Author(s) should kindly revise this.

Response：Thank you for the excellent suggestions. We have restructured the recommendations section following your suggestions. The revised Conclusions and recommendations section are provided below for your convenience.

Line 676-693:

Based on our results, we propose the following policy recommendations: (1) Drive innovation and upgrade industries. Strengthen innovation as a driver for upgrading traditional resource-based industries with green technology. Enhance the technology innovation system, focus on human capital investment, and cultivate independent innovation. (2) Promote interregional resource flow. Improve interregional coordination for efficient resource circulation between urban and rural areas, fostering conditions for industrial upgrades. (3) Enhance pollution control and green tech. Strengthen coordinated pollution control by improving energy efficiency and environmental practices, minimizing spillover effects between cities, and encouraging the use of green technologies. (4) Boost openness and investment. Elevate openness levels through strategic investment attraction in resource-saving and environmentally friendly enterprises, promoting energy-saving technologies and cross-border cooperation. (5) Establish shared development system. Build a shared development system through new infrastructure like "5G" to enhance public services, education, healthcare, and infrastructure investment efficiency. (6) Optimize human capital and transformation. Enhance human capital in resource-based cities through higher education, supporting transformation towards resource-based and innovative industries, and optimizing talent resource allocation.

---

## [Decision Letter · Decision Letter 2]

23 Oct 2023

PONE-D-23-05547R2Spatiotemporal differentiation and analysis of factors influencing high-quality development of resource-based cities: An empirical study based in ChinaPLOS ONE

Dear Dr. 刘,

Thank you for submitting your manuscript to PLOS ONE. After careful consideration, we feel that it has merit but does not fully meet PLOS ONE’s publication criteria as it currently stands. Therefore, we invite you to submit a revised version of the manuscript that addresses the points raised during the review process.

We look forward to receiving your revised manuscript.

Kind regards,

Jing Cheng

Academic Editor

PLOS ONE

Journal Requirements:

Reviewers' comments:

Reviewer's Responses to Questions

**Comments to the Author**

1. If the authors have adequately addressed your comments raised in a previous round of review and you feel that this manuscript is now acceptable for publication, you may indicate that here to bypass the “Comments to the Author” section, enter your conflict of interest statement in the “Confidential to Editor” section, and submit your "Accept" recommendation.

Reviewer #1: All comments have been addressed

Reviewer #3: (No Response)

2. Is the manuscript technically sound, and do the data support the conclusions?

Reviewer #1: Yes

Reviewer #3: Yes

3. Has the statistical analysis been performed appropriately and rigorously? 

Reviewer #1: Yes

Reviewer #3: Yes

4. Have the authors made all data underlying the findings in their manuscript fully available?

Reviewer #1: Yes

Reviewer #3: Yes

5. Is the manuscript presented in an intelligible fashion and written in standard English?

Reviewer #1: Yes

Reviewer #3: Yes

6. Review Comments to the Author

Reviewer #1: This paper utilized the entropy weight TOPSIS method, spatial correlation analysis, and spatial econometric models to explore key factors influencing high-quality development in China's resource-based cities.The author executed a full revision and I recommend its acceptance.

Reviewer #3: 1. The bulleting of the results should be removed.

2. Some paragraphs are too long, especially at the introduction section and must be reduced.

3. The contribution of the study should not be numbered.

4. Table 1 should be well formatted.

5. Remove ‘change’ from line 371.

6. Results paragraph should not be numbered.

7. The manuscript did not have a discussions section (relating findings with existing literature) after the results.

7. PLOS authors have the option to publish the peer review history of their article (what does this mean?). If published, this will include your full peer review and any attached files.

Reviewer #1: No

Reviewer #3: No

---

## [Author Response · Author response to Decision Letter 2]

11 Jan 2024

Dear editor and reviewers,

We greatly appreciate the opportunity to revise our paper, as well as the time and effort that the Academic Editor, Professor Jing Cheng, and the two reviewers have invested in providing constructive and valuable feedback. Following the review team’s suggestions, we have carefully revised our paper as detailed in our point-to-point response below. We are grateful for the feedback and believe that the paper has significantly improved both in content and organization. We hope that you are satisfied with our effort and that you will find the revised manuscript acceptable for publication in the PLOS ONE.

Thank you very much for your assistance with our paper.

1.I can re-upload the ZIP file. When I submitted the revised manuscript, I found that I could not find the place to submit the revised manuscript under attach files. May I ask how I should solve this problem? 

2. The funder was Liu Xiuli, who played an important role in funding acquisition, project management, and review and editing of the writing.

 I am looking forward to hearing from you. Thank you.

---

## [Decision Letter · Decision Letter 3]

26 Jan 2024

PONE-D-23-05547R3Spatiotemporal differentiation and analysis of factors influencing high-quality development of resource-based cities: An empirical study based in ChinaPLOS ONE

Dear Dr. 刘,

Thank you for submitting your manuscript to PLOS ONE. After careful consideration, we feel that it has merit but does not fully meet PLOS ONE’s publication criteria as it currently stands. Therefore, we invite you to submit a revised version of the manuscript that addresses the points raised during the review process.

We look forward to receiving your revised manuscript.

Kind regards,

Jing Cheng

Academic Editor

PLOS ONE

Journal Requirements:

Reviewers' comments:

Reviewer's Responses to Questions

**Comments to the Author**

1. If the authors have adequately addressed your comments raised in a previous round of review and you feel that this manuscript is now acceptable for publication, you may indicate that here to bypass the “Comments to the Author” section, enter your conflict of interest statement in the “Confidential to Editor” section, and submit your "Accept" recommendation.

Reviewer #3: (No Response)

2. Is the manuscript technically sound, and do the data support the conclusions?

Reviewer #3: Yes

3. Has the statistical analysis been performed appropriately and rigorously? 

Reviewer #3: Yes

4. Have the authors made all data underlying the findings in their manuscript fully available?

Reviewer #3: Yes

5. Is the manuscript presented in an intelligible fashion and written in standard English?

Reviewer #3: Yes

6. Review Comments to the Author

Reviewer #3: 1. It will be great if authors could explain the phrase “complex high-quality development”.

2. It is recommended that the numbering of results in the abstract section is removed.

3. I recommend that in the materials and methods section, the write up on data sources precede the variable measurement.

4. In section 4.1.2, the phrase “Glocal autocorrelation” on line 413 should be removed. The proceeding sentence could begin with “This study used …”.

5. It is reommended that the discussion section be separated from the conclusions section. Specifically, the discussion section should be moved beneath the results section.

6. It is important to note that policy ‘implications’ differs from policy ‘recommendations’. It is therefore advised that, given what is presented in the manuscript, authors re-couch the title of the section to ‘recommendations’.

7. PLOS authors have the option to publish the peer review history of their article (what does this mean?). If published, this will include your full peer review and any attached files.

Reviewer #3: No

---

## [Author Response · Author response to Decision Letter 3]

1 Apr 2024

Dear Editors and Reviewers, 

We greatly appreciate the opportunity to revise our paper, as well as the time and effort that the Editor-in-Chief and the reviewer have invested in providing constructive and valuable feedback. Following the review team’s suggestions, we have carefully revised our paper as detailed in our point-to-point response below. We are grateful for the feedback and believe that the paper has significantly improved both in content and organization. We hope that you are satisfied with our effort and that you will find the revised manuscript acceptable for publication in the PLOS ONE.

 For ease of your review, the main changes in the text content of the paper are tracked in BLUE.

Response to Reviewer 1

General comment: The manuscript has been improved but it is recommended that authors work on the following suggestions.

Response: We greatly appreciate the reviewer’s assessment of our work and the helpful comments. The manuscript has been carefully revised according to the reviewer’s comments.

Comment 1: The abstract is lengthy, I recommend that the author(s) focus on the salient points especially for the results.

Response: Thank you for the comment and suggestion. We have rewritten the abstract section to highlight the results section of the paper.

Line 24-40:

Abstract 

Resource-based cities often face problems such as resource scarcity and insufficient electricity to achieve complex high-quality growth. At present, there is relatively little research on the impact on the high-quality development of such cities. To study the key variables that affect the high-quality growth of resource-based cities, we adopt entropy weighted TOPSIS technology, spatial correlation analysis, and spatial econometric models. The main conclusions are as follows: (1) The overall high-quality development of resource-based cities in China is on the rise year by year; The cities with the highest growth rates are those that are mature, rejuvenated, growing, and declining. (2) Resource-based cities have a positive geographical correlation in high-quality development, and different numbers of clusters are displayed by changing the Moran I index score. (3) High quality development is strongly influenced by human capital, urbanization, technological innovation, and global market openness. There are significant differences in the ways in which these variables affect various types of resource-based cities. Policy makers who strive to reduce regional inequality and encourage high-quality growth in resource-based communities may benefit greatly from the insights provided by this study.

Comment 2: In the introduction, line 75, kindly correct this: …, not only to solve their development problems but rather to …, not only to solve their developmental problems.

Response: Thank you for your careful review, we have revised it for your review.

Line 65-69:

The 19th CPC National Congress report pointed out that sufficient support should be provided for the transformation and development of resource-based cities. Therefore, promoting the high-quality development of resource-based cities is necessary, not only to solve their developmental problems but also to implement a national strategy.

Comment 3: The objectives must be well stated.

Response: We thank the reviewer for bringing this to our attention. We have rewritten this paper to describe the objectives of this paper more accurately.

Line78-88:

To fill these gaps, our goal is to examine resource-based cities across China, systematically analyze the spatiotemporal evolution characteristics of high-quality development levels, and the impact mechanisms of different types of resource-based cities. This article takes 113 resource-based cities in China from 2003 to 2018 as samples, uses entropy weighted TOPSIS method and spatial correlation analysis to construct a high-quality development evaluation system, and studies the spatiotemporal evolution trends of high-quality development in different types of cities. Meanwhile, using spatial econometric models, this study explores the roles of industrial structure, technological innovation, human capital, urbanization, environmental regulation, openness, and infrastructure in high-quality development. Finally, based on empirical results, high-quality development suggestions were proposed.

Comment 4: The paragraph on the contribution of the study is not clear (line 95-106). Are the author(s) referring to the significance of the study?

Response: Thanks for kindly reminding us to clarify this point. The contribution of this paragraph refers to the significance of this study. We have rewritten the contribution of this study to make it clearer and more focused.

Line 89-102:

The contribution of this study is mainly reflected in three aspects: firstly, this study developed an inclusive and widely applicable evaluation index system for the high-quality development of resource-based cities. This innovative system is based on the “Five Development Concepts” and fills an important gap in the current research field, providing a comprehensive framework for evaluation. Secondly, by incorporating most resource-based cities in China into the research sample, this study provides a comprehensive analysis with practical significance covering a wider range of regions. This helps to improve the applicability of research results. Finally, this study selected relevant influencing factors through scientific and targeted methods, thus filling the research gap in the factors affecting the high-quality development of resource-based cities. Through this comprehensive analysis, this study reveals the challenges faced by resource-based cities in the process of high-quality development and provides wise solutions, which helps decision-makers better understand the essence of high-quality development in resource-based cities.

Comment 5: The section on indicator system should be part of the literature while the specific indicators used and their ranges could be in the methods section.

Response: We agree with the reviewer’s point. Based on your suggestions, we present the indicator system in the literature review section and the specific indicators in the methodology section.

Line 179-187:

3.1.1 Measurement of high-quality development level of resource-based cities

(1) Indicator system. This study is based on “five development concepts”. It also draws on relevant research on sustainable development[39], human development evaluation[40], high-quality development in China in the new era[41-43], and resource-based city transformation[44]. According to the special objective situation of resource-based cities[44], this study establishes a high-quality development index system for resource-based cities. Six subsystem evaluations of economy, innovation, coordination, greenness, openness, and sharing are also constructed. Using the entropy weight method, the weight of each evaluation index is determined (Table 1).

Comment 6: Author(s) should check the version of ArcGIS used… it should be ArcGIS 10.2. This should be mentioned in the methods section and not the results. Stata is also mentioned in the results section but it would be more appropriate to mention the analytical software in the methods.

Response: Thank you for this excellent suggestion and for motivating us to push our work further. We have checked the version of ArcGIS as 10.2. and we have mentioned the software we are using in the methodology section.

Line 246-249:

(2) Local spatial autocorrelation model. As the global spatial autocorrelation Moran's I index cannot reflect the heterogeneity between regions, the local Moran's I index, namely the local index of spatial correlation (LISA), is introduced to reveal the relationship between resource-based and neighboring cities using ArcGIS 10.2 software.

Line 259-263:

3.1.3 Spatial Econometric Model

The high-quality development of resource-based cities is not only affected by these factors but also by the development of neighboring cities. Therefore, this study uses a spatial econometric model to conduct an empirical test of the factors affecting the high-quality development of resource-based cities using Stata 15.0 software.

Line 323-326:

Individual missing data are estimated using linear interpolation, and the data are logarithmically processed to eliminate the effect of heteroscedasticity. The data processing software used in this paper is Stata15.0. The descriptive statistics for the variables are presented in Table 2.

Comment 7: The version of ArcGIS used is not consistent. At a point, author(s) state it is 10.2 and in 10.3 at other points. Author(s) should kindly clarify this.

Response: We thank the reviewers for drawing our attention to this point. We have checked the version of the ArcGIS version for 10.2 and revised it throughout.

Comment 8: It is inappropriate to cite other scholarly works at the recommendations section. Author(s) should kindly revise this.

Response: Thanks to the reviewer for this careful comment. We have eliminated references to other scholars in the recommendations section.

Line 671-689:

6. Policy implications

Based on our results, we propose the following policy recommendations: (1) Drive innovation and upgrade industries. Strengthen innovation as a driver for upgrading traditional resource-based industries with green technology. Enhance the technology innovation system, focus on human capital investment, and cultivate independent innovation. (2) Promote interregional resource flow. Improve interregional coordination for efficient resource circulation between urban and rural areas, fostering conditions for industrial upgrades. (3) Enhance pollution control and green tech. Strengthen coordinated pollution control by improving energy efficiency and environmental practices, minimizing spillover effects between cities, and encouraging the use of green technologies. (4) Boost openness and investment. Elevate openness levels through strategic investment attraction in resource-saving and environmentally friendly enterprises, promoting energy-saving technologies and cross-border cooperation. (5) Establish a shared development system. Build a shared development system through new infrastructure like "5G" to enhance public services, education, healthcare, and infrastructure investment efficiency. (6) Optimize human capital and transformation. Enhance human capital in resource-based cities through higher education, supporting transformation towards resource-based and innovative industries, and optimizing talent resource allocation.

Again, we highly appreciate the opportunity to revise and resubmit the manuscript again as well as your comments and suggestions on it. They have helped us improve the manuscript significantly. We hope we have answered all of your questions and have revised the manuscript to your satisfaction.

---

## [Editor Report · Decision Letter 4]

14 May 2024

Spatiotemporal differentiation and analysis of factors influencing high-quality development of resource-based cities: An empirical study based in China

PONE-D-23-05547R4

Dear Dr. 刘,

We’re pleased to inform you that your manuscript has been judged scientifically suitable for publication and will be formally accepted for publication once it meets all outstanding technical requirements.

Kind regards,

Chenxi Li

Academic Editor

PLOS ONE
---

## [Editor Report · Acceptance letter]

21 May 2024

PONE-D-23-05547R4 

PLOS ONE

Dear Dr. Liu, 

I'm pleased to inform you that your manuscript has been deemed suitable for publication in PLOS ONE. Congratulations! Your manuscript is now being handed over to our production team.

Kind regards, 

on behalf of

Dr. Chenxi Li 

Academic Editor

PLOS ONE